# Suppressing Sidelobes in Metasurface-Based Antennas Using a Cross-Entropy Method Variant and Full Wave Electromagnetic Simulations

**Khushboo Singh** \*[ID] **and Karu Esselle** [ID]

School of Electrical and Data Engineering, University of Technology Sydney, Ultimo, NSW 2007, Australia; karu.esselle@uts.edu.au
\* Correspondence: khushboo.singh@uts.edu.au

**Abstract:** Managing sidelobe levels (SLLs) in metasurface-driven beam-steering antennas poses a significant challenge due to intrinsic factors leading to grating lobes. Our proposed method employs an equivalent model to efficiently optimize large periodic metasurfaces. This model predicts complete metasurface performance, accounting for mutual coupling between patches. We introduce an evolutionary optimization algorithm based on the cross-entropy (CE) method to enhance PGM-based beam-steering antennas and suppress sidelobes. Two strategies are employed: the first is to optimize the patch dimensions for a sidelobe-free pattern, and the second is to maintain the PGM dimensions while optimizing the feed array amplitudes. Both strategies effectively suppress sidelobes, offering insights into the CE method's applicability and effectiveness for CPU-intensive electromagnetic optimization challenges. The proposed CE method variant retains its simplicity while improving monitoring capabilities, addressing this limitation. Smaller generations yield better improvements per evaluation. The uniqueness of the proposed optimization strategy lies in its utilization of an equivalent 1D metasurface model for optimization that not only considers the mutual coupling between identical unit cells along the y-direction within a complete metasurface but also takes into account the distinct cells along the x-direction. Moreover, the 1D metasurface model incorporates the influence of edge effects along the x-direction.

**Keywords:** metasurface; metamaterial; optimization; phase-shifting surface; beam steering; beam scanning; phase-gradient surface; artificial surface; phase-correcting structure; cross-entropy method; sidelobes; grating lobes





## 1. Introduction

Metasurfaces are expansive electromagnetic structures designed by meticulously arranging sub-wavelength inclusions. These arrangements endow these structures with an extraordinary capability for precise manipulation of electromagnetic wave attributes, including amplitude, frequency, phase, and polarization [1,2]. This refined control over signal propagation characteristics holds significant promise across diverse wireless communication applications.

High-gain antennas are frequently integrated with phase-gradient metasurfaces (PGMs) to achieve precise control over antenna beam steering within a wide conical range [3]. This combination proves particularly valuable for establishing network connectivity via low/medium-earth-orbit (LEO/MEO) satellites. These configurations address the demands of satellite communications, specifically targeting scenarios involving on-the-pause or on-the-move operations. Their applications extend to providing essential connectivity in remote areas or dynamic environments, such as flights, ships, and trains, where conventional terrestrial networks encounter limitations.

The phased-gradient metasurfaces are constructed using repeating supercells, each comprising a collection of unique unit cells. These individual unit cells are associated

with specific phase delays, creating a comprehensive phase spectrum of 360 degrees. The arrangement of these unit cells within a supercell ensures a consistent phase difference between adjacent cells across the entire metasurface. Given the cyclical properties of phase values (after every 360°), the supercells exhibit repetitive patterns across the aperture of the PGM [4]. These electrically large-aperture PGMs can be conceptualized as versatile devices akin to generalized reflectors or refractors, which inadvertently generate unintended diffraction orders, leading to the formation of periodic lobes [5–9].

The presence of spurious sidelobes and grating lobes can lead to signal leakage, representing significant challenges in the context of beam-scanning antennas. Achieving optimal steering performance requires an efficient metasurface with a tailored amplitude and phase response corresponding to lower sidelobes and grating lobes. Effective control over these undesired lobes can be achieved by skillfully managing the arrangement of elements within the periodically repeating supercell, as well as optimizing the dimensions of the metallic features within the PGMs. This comprehensive approach contributes to enhanced antenna performance and improved beam-steering capabilities [10,11].

While optimization techniques have gained traction in electromagnetic (EM) engineering, evolutionary optimization methods have been sparingly employed in the optimization of metasurfaces [12,13]. Beam-steering antenna systems based on metasurfaces often grapple with the challenge of excessive sidelobes within their radiation patterns. These sidelobes are undesirable as they lead to signal leakage and power loss. Furthermore, for applications involving satellites on the move (SOTMs), stringent criteria regarding directivity, bandwidth, and sidelobes must be met. In this work, we propose a variant of the cross-entropy method and implement it to mitigate the sidelobe levels (SLLs) within the radiation pattern of a metasurface-based beam-steering antenna.

The domain of global evolutionary optimization methods encompasses a diverse array of techniques. Notably, Genetic Algorithms (GAs), Particle Swarm Optimization (PSO), and Covariance Matrix Adaptation Evolution Strategies (CMA-ES) emerge as prominent contenders [14]. These methodologies have gained widespread traction in addressing intricate optimization challenges within the field of electromagnetics in recent years. It is imperative to not only establish but also substantiate the efficacy of the cross-entropy method in achieving performance parity with or possibly surpassing these firmly established techniques. To validate the capabilities of the cross-entropy method, we draw the reader's attention to the investigations carried out in [14], where a comparative assessment of the cross-entropy method against PSO and CMA-ES was conducted. The findings elucidated that while all three methods ultimately converged toward similar solutions, the cross-entropy method demonstrated a pronounced advantage in terms of computational efficiency. This empirical evidence strongly supports the assertion that the cross-entropy method is a faster option for electromagnetic optimizations.

In the traditional cross-entropy (CE) approach, each candidate is evaluated against others within the same generation to determine its elite status, with no consideration for the distribution of candidates in prior generations. Similar to the conventional CE method, this variant also necessitates a memory capacity of size $N_q$ in every iteration. However, after the initialization of the queue, which entails $N_q$ performance evaluations, subsequent iterations only require a single performance evaluation for each new sample. These new samples are then compared against the most recent $N_q$ candidates in the queue, allowing for an efficient and dynamic optimization process. Through the prompt update of the sampling distribution following each candidate evaluation, the algorithm efficiently navigates the design space. This approach obviates the need for exhaustive evaluations across the entire population, rendering it well-suited for optimizing intricate metasurface designs. This method employs continuous monitoring and adaptive adjustments in response to the algorithm's search progress. Furthermore, the variant with a population size of $N = 1$ presents a close approximation to a continuous process, potentially offering greater analytical clarity compared to discrete methods.

In contrast to the previously mentioned related article [8], which is essentially a very concise study, this manuscript provides an intricate exploration of the optimization process, along with a justification for selecting specific probability distribution functions over other options. The detailed clarification of the cross-entropy method variant contributes a sturdy conceptual framework that can be applied to a variety of complex problems. Notably, there has been no prior comprehensive analysis of this particular variant of the CE method in the context of electromagnetic optimization, making it a pivotal component. Furthermore, it is important to note that the optimization results presented in [8] were based on a periodic feed distribution in contrast to our approach, where we employ a radial symmetry in the feed amplitude distribution. This radial symmetry aligns more closely with the ideal scenario for feed tapering, such as the Taylor or Chebyshev taper distributions. In this research, we also conduct a comparative analysis of two distinct optimization strategies that can be implemented either separately or in combination. The objective of this analysis is to enhance overall performance by reducing the sidelobes and grating lobes in a metasurface-based beam-steering antenna system.

## 2. The Cross-Entropy Method

The concept of the cross-entropy (CE) method originated in 1997 [15,16]. In 1999, Rubinstein refined and applied the CE method specifically for combinatorial optimization [17], conducting extensive testing on challenging benchmark problems [18–21]. One outstanding feature of the CE method is its ability to rapidly converge toward the optimal or nearly optimal solution. The rules for updating the parameterized probability distribution functions (PDFs) are straightforward to implement and have a solid theoretical basis rooted in information theory, providing a strong theoretical justification [22]. To estimate the optimal probability distribution that generates globally optimal solutions, the CE method follows a two-step iterative process [23,24]:

1.  Generate a set of candidate solutions from a pre-defined parameterized probability distribution.
2.  Adaptively update the parameters of this probability distribution using the information extracted from the current elite candidate solutions. This adaptive update aims to guide the search toward the global optimum by minimizing the *cross-entropy* (or Kullback–Leibler divergence) between two sampling distributions.

The CE method is a highly versatile global stochastic learning optimization approach capable of handling both combinatorial and continuous multi-extremal, multi-objective optimization challenges. One of its distinguishing characteristics is that it operates on a parameterized probability distribution rather than directly on samples within the candidate population. The method employs an iterative process, updating the sampling distribution while minimizing the cross-entropy between the empirical distribution of the current elite sub-population and the sampling distribution of the subsequent iteration. This iterative refinement continues until the cross-entropy reaches zero, indicating that the two distributions have become identical, thereby achieving the global best solution.

The CE optimization method follows a model-based search framework. In this approach, feasible solutions are derived from a parameterized probability distribution function (PDF), which is continually updated based on the elite candidates identified in the previous iteration. Probability distributions are commonly referred to as "models" in the literature [25]. An elementary schematic diagram illustrating the CE search framework is presented in Figure 1. In essence, the cross-entropy (CE) method relies on a predefined PDF (model). The overall process can be concisely described as an iterative sequence encompassing the definition of a model, the sampling of data from that model, learning from the sampled data, and subsequently updating the initial model.

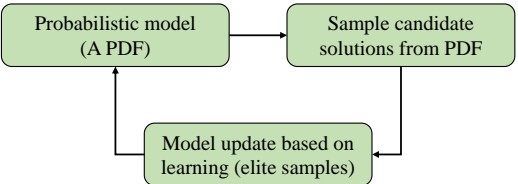

**Figure 1.** Model-based search framework for CE method.

### 3. Cross-Entropy Method Variant

The CE method adopts a population-based strategy, where it evaluates all samples within a generation (population size N) before updating the sampling distribution. This process is referred to as the *batch processing* of the population. In the traditional CE algorithm, candidates are selected only after the entire generation evaluation episode is completed. However, this approach becomes problematic when dealing with computationally expensive and time/memory-consuming problems. To address this issue, the proposed variant of the CE method performs updates immediately after each candidate is evaluated, eliminating the need to wait for the entire generation's evaluation to be completed.

To implement the variant of the CE method, we define the problem-based performance metric $F(x)$ over the solution space $X$, where $F(x) \in \mathbb{R}$ for all $x \in X$. The goal is to estimate the parameters of the optimal probability distribution function (PDF) that generates the globally optimal or near-optimal solution. Similar to the classical CE method, this variant starts with an initial PDF and progressively constructs a sequence of sampling distributions, focusing increasingly on a small neighborhood around the optimal solution. Finite-dimensional PDFs are used, and the choice of these PDFs depends on the nature of the design parameters associated with the optimization problem, that is, whether they are continuous, discrete, or mixed parameters. Selecting a PDF that accurately models the problem's structure is crucial. The optimization problems solved by the CE method are often multi-dimensional and may involve constraints. The parameters in such problems can be either continuous, discrete, or a combination of both (mixed parameters). To ensure proper sampling of the candidate solutions, the PDF should appropriately support the range and type of parameters involved [26].

The first step is to sample candidates from a carefully chosen parameterized PDF. The continuously constrained parameters are usually randomly sampled, using either *acceptance rejection* or *Gibbs sampling*. The inversion method is often used to sample from a discrete distribution [27]. The second step is called parameter estimation. We use the maximum likelihood estimation (MLE) method [28] to estimate the parameters of a sampling distribution. In information theory, a natural way to estimate the parameters of the new PDF is to update the parameters of the current PDF by minimizing the *Kullback–Leibler divergence* (a measure of misfit between two distributions), as expressed in (1) for continuous distributions and (2) for discrete distributions:

$$D_{KL}(p||q) = \int_X p(x) \ln \frac{p(x)}{q(x)} dx,$$ (1)

$$D_{KL}(p||q) = \sum_{x \in X} p(x) \ln \frac{p(x)}{q(x)} dx,$$ (2)

or, analogously, by minimizing the *cross-entropy*, as expressed in (3) for continuous distributions and (4) for discrete distributions:

$$H(p,q) = - \int_X p(x) \ln q(x) dx,$$ (3)

$$H(p,q) = - \sum_{x \in X} p(x) \ln q(x),$$ (4)

between two PDFs, $p(x)$ and $q(x)$, over $X : x \in X$. When any of the equations mentioned above, such as $p(x) = q(x)$, equal zero, it indicates a perfect estimate of PDF parameters, resulting in both distributions becoming identical. The maximum likelihood estimation (MLE) and cross-entropy (or Kullback–Liebler divergence) minimization are essentially equivalent approaches, and one can be readily derived from the other [25]. Consequently, either of these methods can be employed for optimization using the CE method.

We now introduce the necessary user-defined parameters for the CE method variant. The input parameters include the queue size $N_q$, elite sub-population size $N_{el}$, initial sampling distribution parameters, and smoothing parameter $\alpha$. The algorithm is initialized with a random start [29]. In each iteration, a candidate is sampled from the initial distribution, and its fitness function is evaluated. The sampled candidate, along with its fitness function value, is stored in the queue. This sampling and storing process is repeated $N_q$ times until the queue reaches its capacity. Once the queue is full, the candidates are sorted based on their fitness values, either in ascending order if the optimizer aims to minimize the fitness function, or in descending order if the objective is to maximize it. The first $N_{el}$ candidates are then selected as the *elite* candidates, where $N_{el}$ is calculated as $\rho \times N$ with $0.01 \leq \rho \leq 0.1$. Additionally, the best candidates and their corresponding fitness values are recorded. The optimizer then checks the convergence or stopping criterion. If the criterion is not met, the oldest candidate is dropped from the queue and the iteration continues. Each candidate sampled in the following iterations will be compared against the last $N_q$ candidates, and the decision regarding the candidate being elite can be made instantly as opposed to the traditional method, where a whole population of candidates has to be evaluated before the algorithm can decide if the candidate is elite.

In each subsequent iteration, new candidates are sampled from an updated probability distribution function (PDF) and evaluated until the stopping criterion is satisfied. The parameters of the current sampling PDF are updated based on the elite sub-population to define a new sampling distribution for the next iteration. As previously mentioned, this updating process can be achieved by minimizing the cross-entropy or utilizing the concept of the maximum likelihood estimation (MLE) to find the best fit between the empirical distribution of elite candidates and the new sampling distribution. To strike a balance between exploitation and exploration and prevent premature convergence, the algorithm incorporates a smoothing parameter $\alpha$. This parameter ensures a satisfactory trade-off between intensification (exploitation) and diversification (exploration). The $\alpha$ smoothing is integrated into the updating rules as shown in the following equation:

$$v_t^s = \alpha v_t + (1 - \alpha)v_{t-1}, \tag{5}$$

where $v_t$ is the current generation PDF parameter vector, $v_{t-1}$ is the PDF parameter vector from the previous generation, and $v_t^s$ is the smoothed parameter vector. The smoothing parameter $\alpha$ lies between 0 and 1. A higher value of $\alpha$ (closer to 1) results in faster convergence, whereas a lower value of $\alpha$ slows the convergence. For high-dimensional problems (more than five), fast convergence usually results in a decreased chance of finding the global best solution. An alternate smoothing procedure known as *dynamic* or $\beta$ *smoothing* is followed in cases where the $\alpha$ smoothing results in a quick sub-optimal convergence [23].

There are several ways to define the termination criterion. Some of the popular stopping conditions are as follows:

1.   When the elite sub-population consists of identical or very similar results.
2.   When a specified number of maximum iterations are completed.
3.   When the distance between the best-found solution and the target result becomes negligible.

An elaborate discussion on the stopping criterion is available in [30]. Several modifications of the CE method, such as the *Variance-Injection Method* and *Fully Adaptive CE method*, are described in [23]. Improved CE methods and their online variants can be found in [31]. The iterative steps of this CE method variant are summarized in Algorithm 1.

---

**Algorithm 1** Pseudo-code for CE method variant.

---

    *Inputs:*
    $f : P \rightarrow \mathbb{R}$ , objective function to be minimized
    *Operating Parameters:*
    $N_q$, queue size
    $N_{el} < N_q$ , number of elite candidates in the queue
    $\alpha$, smoothing parameter
    $\theta_0$, initial sampling distribution
    $\{S(\theta)\}$, parameterized family of sampling distributions
    *Initialization:*
    $\theta := \theta_0$
    **for** $k = 1$ to $N_q - 1$ **do**
        Draw $x_k$ as a random variable from the distribution $S(\theta)$
        Evaluate $f(x_k)$
    **end for**
    $k \leftarrow k + 1$
    *Main Loop:*
    **while** Stopping Criterion Not True **do**
        Draw $x_k$ as a random variable from the distribution $S(\theta)$
        Evaluate $f(x_k)$
        $Q := (x_{k-N_q+1}, x_{k-N_q+2}, \ldots, x_k)$
        *# Sorting*
        Order the elements of $Q$ as $\tilde{Q} := \{\tilde{x}_1, \tilde{x}_2, \ldots, \tilde{x}_{N_q}\}$ such that $f(\tilde{x}_1) \leq f(\tilde{x}_2) \leq \ldots \leq$
    $f(\tilde{x}_{N_q})$.
        *# Recording the best values*
        $(x_{best}, f_{best}) := (\tilde{x}_1, f(\tilde{x}_1))$
        *# Elite population*
        $X_{el} := (\tilde{x}_1, \tilde{x}_2, \ldots, \tilde{x}_{N_{el}})$
        *# Minimizing cross-entropy between new sampling distribution and elite population*
        Calculate $\tilde{\theta}$ that maximizes the agreement between $S(\tilde{\theta})$ and $X_{el}$.
        *# Parameter Update*
        $\theta \leftarrow \alpha\tilde{\theta} + (1 - \alpha)\theta$
        $k \leftarrow k + 1$
    **end while**
    *Outputs:*
    $(x_{best}, f_{best})$

---

## 4. Periodic Metasurface Optimization Methodology

A periodic phase-gradient metasurface is shown in Figure 2. These metasurfaces entail an electrically expansive planar configuration consisting of repeating supercells arranged along both the *x*- and *y*-axes. This arrangement orchestrates a progressive phase modulation in the electric field at the output, facilitating controlled beam tilting for the antenna's operation.

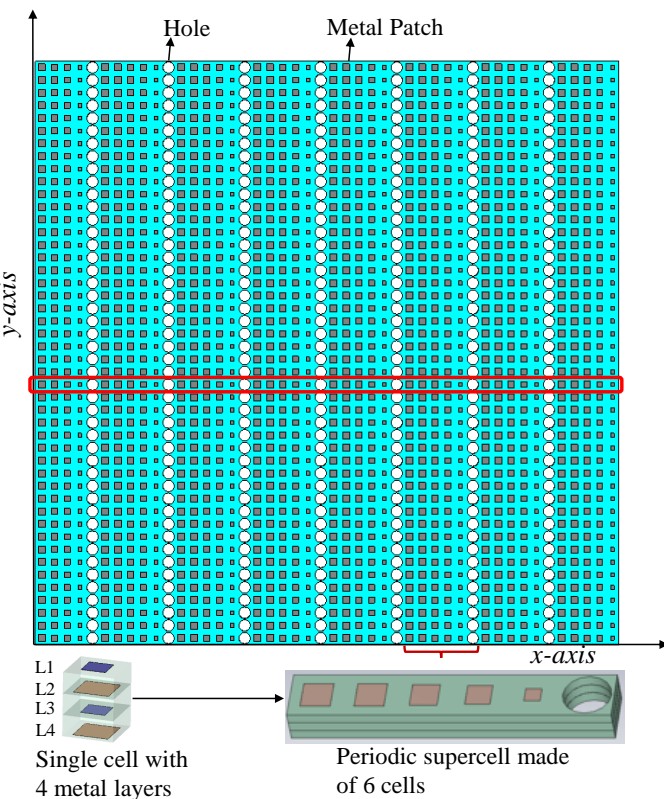

**Figure 2.** Complete architecture of a PGM, with a supercell and a single-cell internal composition added at the bottom for clarity.

Because of the absence of a precise analytical model for metasurfaces, their effective optimization relies on employing a comprehensive electromagnetic simulation model.

Metasurfaces typically have electrically large apertures and a significant number of small features. As a result, simulating them becomes computationally demanding and impractical, particularly when considering population-based global optimization using conventional approaches. To mitigate the computational complexity, a more straightforward equivalent model (illustrated in Figure 3) has been proposed.

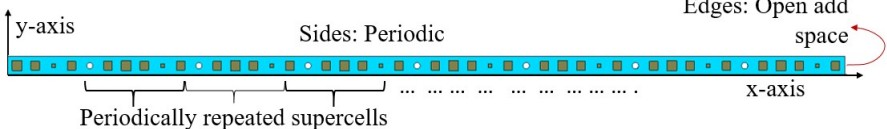

**Figure 3.** Simplified equivalent model of a metasurface, with boundary conditions specified to mimic a full 2D metasurface.

We construct an equivalent 1D metasurface model, which corresponds to the highlighted section in red in Figure 2. It has the same $x$-axis dimension (L = 345 mm) as the original metasurface aperture and a $y$-axis dimension (W = 7.5 mm) as small as the dimension of the constituent unit cell. Appropriate boundary conditions are assigned to the 1D metasurface model in the CST-MWS time-domain solver to emulate the response of a full metasurface ("E(t) = 0" along *the y*-axis and the "Open Add Space" boundary condition along *the x*-axis, as depicted in Figure 3. Despite the simplification, the structure retains a high mesh complexity (exceeding 800,000 hexahedrons), necessitating the avoidance of batch processing.

As a solution, we adopt the modified version of the CE (cross-entropy) method to optimize these printed metasurfaces. Our focus is on achieving sidelobe suppression and obtaining a radiation pattern that conforms to the FCC mask (25.209) for the Ka-band.

The algorithm for this customized CE method is implemented in MATLAB and linked with CST-MWS using a VBA–macro code. Through the implementation of two illustrative examples, we showcase the capability of this generalized CE approach to effectively manage computationally demanding electromagnetic challenges. This emphasizes its potency as a valuable tool for optimizing intricate metasurface designs.

Unlike many other electromagnetic optimizations based on analytical approaches, which often rely on various assumptions, our optimization methods stand out by taking into account the coupling effect between adjacent non-identical unit cells along the x-direction, as well as the identical unit cells along the y-direction in this simplified equivalent model. In our optimization of the 1D metasurface model, we consider all the unit cells present in the actual finite-sized metasurface along the x-direction. This is because they only exhibit supercell periodicity along this direction and are essentially locally non-periodic. However, along the y-direction, metasurfaces are periodic at the unit-cell level, so we model the mutual coupling effect using periodic boundary conditions. The actual finite-sized metasurface effectively repeats the unit cells a sufficient number of times to mimic infinite periodicity in this direction. Another noteworthy feature of our approach is the inclusion of edge effects along the x-direction. This is a departure from other optimization approaches presented in [1,9], where such edge effects were not considered.

### 4.1. Optimizing the Patch Dimension in a PGM to Control SLLs

By employing the outlined CE methodology, we embark on the optimization of the metasurface design with the goal of reducing the sidelobe levels present in the radiation pattern of metasurface-based antennas. As shown in Figure 2, each supercell of a PGM is composed of an array of non-identical phase-transforming cells (PTCs), wherein each PTC corresponds to a specific transmission phase and exhibits a substantial level of transmission magnitude. These PTCs serve as the fundamental building blocks of a phased-gradient metasurface (PGM), governing the spatial phase variation of the electric field as it traverses through them. Within this specific PGM design, two distinct types of phase-transforming cells (PTCs) are integrated, each possessing a length equivalent to $\lambda_0/2$.

The Type-I cell comprises four square metal patches enveloped by three dielectric layers. These patches, denoted as $L1$, $L2$, $L3$, and $L4$, are dimensioned such that $L1 = L3$ and $L2 = L4$. Conversely, the Type-II cell is characterized by three stacked dielectric layers containing a through-hole. The need for a multi-layer unit element is driven by the essential requirement of achieving a $360°$ phase range, a crucial aspect in the design of a phase-gradient metasurface (PGM), as elucidated in [1,2]. The Taconic TLY-5 dielectric material featuring a permittivity of $\epsilon_r = 2.2$ is employed for both PTC variants. Utilizing a unit cell with through holes extends the attainable phase range from a particular unit cell. For a more in-depth explanation, interested readers are referred to [2], where a detailed elucidation on the design methodology for such PGMs is provided, offering enhanced clarity on the subject.

The overall structure is created by repeating a periodic supercell, which consists of five Type-I cells and one Type-II cell. This supercell has dimensions of $l = \lambda/2 \times 6 = 3\lambda$. To cover the complete phase range from $0°$ to $360°$, the dimensions of the square metal patches and the openings within the PTCs are modified accordingly. Within the supercell, the phase delay between adjacent cells is set at $60°$ or $\pi/3$ radians. According to array theory, the beam tilt attained through the 1D array can be mathematically expressed using the following equation [32]:

$$\sin\theta - \sin\theta_i = \frac{\Delta\phi}{d}\frac{\lambda_0}{2\pi},\tag{6}$$

In this context, the symbol $d$ stands for the spacing between elements within the array, $\Delta\phi$ signifies the incremental phase shift contributed by each array element in radians, $\lambda_0$ denotes the wavelength in free space, $\theta_i$ represents the angle of incidence concerning the surface normal, and $\theta$ indicates the desired beam tilt concerning the central radiation

direction. Consequently, under the conditions of normal incidence, specifically when $d = \lambda_0/2$ and $\Delta\phi = \pi/3$, the metasurface achieves a beam tilt of $20°$, as elucidated by (6).

The optimization involves the design variables depicted in Figure 4. Within the metasurface, specific parameters are fixed, including the aperture size ($23\lambda_0 \times 23\lambda_0$), cell size ($d = \lambda_0/2$), and the permittivity of the dielectric substrate ($\varepsilon_r = 2.2$). In the context of the 1D metasurface model, uniform plane-wave illumination is applied through a waveguide port positioned one wavelength below the metasurface. As the supercells are essentially replicated along the x-direction, the design dimensions of the metallic strips and holes within the supercell also follow this repetition across the metasurface. Consequently, the side lengths of the first (top)- and third-layer metal patches ($a1, a2, a3, a4, a5$), as well as the second- and fourth (bottom)-layer metal patches ($b1, b2, b3, b4, b5$), for five consecutive cells, along with the radius ($r$) of the hole within the supercell (as depicted in Figure 4), collectively form an 11-parameter design vector, as defined in Equation (7):

$$x = (a1, a2, a3, a4, a5, b1, b2, b3, b4, b5, r). \tag{7}$$

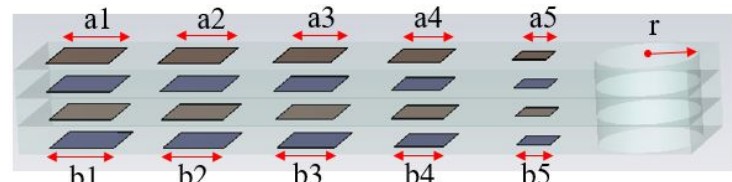

**Figure 4.** The metasurface supercell, showing the design parameters used in metasurface optimization.

The optimization problem for the metasurface design is formulated with a set of inequality constraints, as represented in Equation (8).

$$\begin{cases} 0.05 \text{ mm} \leq \text{a1, a2, a3, a4, a5, b1, b2, b3, b4, b5} \leq 7 \text{ mm} \\ 0.05 \text{ mm} \leq (r) \leq 3.5 \text{ mm}. \end{cases} \tag{8}$$

These constraints are derived from prior knowledge, indicating that the patch dimensions within the unit cells are varied within the range of 0.05 mm to 7 mm. Similarly, for the unit cell with a through hole, the radius is varied within the range of 0.05 mm to 0.35 mm. It is important to note that these parameter limits have been set with careful consideration of the maximum side length of the square unit cell, which is 7.5 mm, to prevent any overlap issues and ensure that the optimized metasurface design remains physically feasible and practical for fabricating a realistic and achievable design.

### 4.1.1. Algorithm Implementation

The optimization procedure described for the metasurface represents a classical instance of continuous optimization within a constrained search space. Given the strong mutual coupling among the metallic patches, there is a significant correlation among the parameters within the distinguished sub-population. To effectively handle this correlation, we employ a multivariate Gaussian distribution to sample the 11 optimization parameters. This approach adeptly captures the correlation of continuous parameter values by refining the sampling distribution for each subsequent generation. The distribution parameters, namely the mean $\mu$ and the covariance $\Sigma$, are employed such that $x \sim \mathcal{N}(\mu, \Sigma)$. The probability density function (PDF) of a multivariate Gaussian distribution is expressed as:

$$\mathcal{N}(x|\mu, \Sigma) = \frac{1}{\sqrt{(2\pi)^d |\Sigma|}} exp\left(-\frac{1}{2}(x - \mu)^T \Sigma^{-1}(x - \mu)\right), \tag{9}$$

where $x$ is a vector of random variables of dimension $d$, $\mu$ is a ($1 \times d$) vector of means, $\Sigma$ is the covariance matrix of dimension ($d \times d$), and T stands for transpose. To uphold the boundaries of the solution space, a sigmoid transformation is implemented, which maps

$\mathbb{R}^{11}$ to a viable rectangle, as detailed in [33]. This transformation ensures that the obtained samples fall within the predetermined limits. The initial distribution is characterized by a mean $\mu = 0$ and a covariance $\Sigma = I$, with $I$ signifying an identity matrix. These parameters are subsequently adjusted using information from the elite sub-population following each evaluation.

The primary objective involves the minimization of the sidelobe levels (SLLs) within the directivity pattern of the metasurface-based antenna, operating at a frequency of 20 GHz. Central to this endeavor is the precise definition of a fitness function, serving as a singular performance metric for the effective assessment of potential solutions. A novel approach has been introduced for formulating this fitness function, with a modified aim of ensuring adherence to the prescribed FCC (25.209) mask specific to the *Ka*-band.

The *mask* function, as depicted in Equation (10), establishes the upper limit for the desired gain (in **dBi**) across various elevation angles $\theta$, ranging from $-180°$ to $180°$ when the peak radiation aligns with the broadside direction. Each $\theta$ angle corresponds to an allowable gain level. Should the radiation beam be steered by $\delta$ degrees, the mask adjusts to align with the peak radiation direction of the beam. Consequently, the expression for the *mask* function when dealing with a tilted beam transforms into ($|\theta| - \delta$).

$$Mask = \begin{cases} 24.6 \ \mathbf{dBi}, & 1.5° < |\theta| \leq 7° \\ 7.8 \ \mathbf{dBi}, & 7° < |\theta| \leq 9.2° \\ 8 \ \mathbf{dBi}, & 9.2° < |\theta| \leq 19.1° \\ 0 \ \mathbf{dBi}, & 19.1° < |\theta| \leq 180° \end{cases} \tag{10}$$

The algorithm aims to minimize the fitness function (*FF*), which is formulated as:

$$FF = \sum_{\theta=-180}^{180} \left( min(0, (Mask(\theta) - Directivity(\theta))) \right)^2 \tag{11}$$

The *FF* value is computed across all elevation angles $\theta$, ranging from $-180°$ to $180°$ in increments of $1°$. Equation (11) ensures that the squared difference contributes to the *FF* value only when the directivity pattern lies above the mask, whereas no contribution is made when the pattern falls below the mask. The primary objective of the optimization process is to minimize the *FF*, thereby effectively diminishing the discrepancy between the mask and the directivity pattern whenever the pattern violates the mask's requirements. The user-defined parameters governing the algorithm for the proposed CE method variant in the context of metasurface optimization are concisely summarized in Table 1.

**Table 1.** The CE method parameters for optimization of PGMs.

| Queue Size $N$ | Elite Sub-Population Size $N_{el}$ | Smoothing $\alpha_S$ |
| --- | --- | --- |
| 110 | 11 | 0.2 |

The phased-gradient metasurface exhibits inherent local non-periodicity and contains sub-wavelength metallic features. As a result, conventional techniques, such as transmission-line modeling and unit cell optimization [34,35], are unsuitable for comprehensively analyzing its intricate electromagnetic characteristics. To accurately investigate the complex electromagnetic behavior, we adopt a full-wave electromagnetic simulation-driven optimization strategy. This approach serves to predict the far-field radiation pattern of the beam-steering system based on the phased-gradient metasurface (PGM). To facilitate this pursuit, we employ the cross-entropy (CE) algorithm, which is implemented in MATLAB. This algorithm is seamlessly interfaced with CST MWS (Microwave Studio) through a macro code. This linkage establishes a dual-channel connection between MATLAB and

CST, enabling smooth communication and seamless integration between the optimization algorithm and the detailed full-wave electromagnetic simulations.

### 4.1.2. Optimization Results

Each evaluation conducted within the CST MWS time-domain solver necessitated approximately 28 min, and utilized an Intel Core i7-6700 CPU clocked at 3.4 GHz with 64 GB RAM. The entirety of the optimization procedure, which encompassed 959 function evaluations, spanned a cumulative duration of 447 h and 48 min. To effectively account for alterations in patch geometry during each function evaluation, an adaptive meshing technique was implemented. The optimization cycle persisted until the maximum deviation of patch dimensions in an elite sample from the mean of all corresponding elite patch dimension samples within the queue became less than 0.1 mm. As illustrated in Figure 5, the convergence curve reveals that the optimal solution was achieved at the 959th function evaluation. Throughout the optimization progression, once the queue became populated with $N_q$ samples, the average fitness of the queue was graphed in parallel with the best fitness attained up until that juncture. Here, "best fitness" denotes the most favorable outcome discovered in the queue until that instance, whereas "average fitness" signifies the mean of the fitness values for all candidates within the queue at each function evaluation. As depicted in Figure 5, we can observe a gradual decrease in the average fitness function with an increasing number of function evaluations.

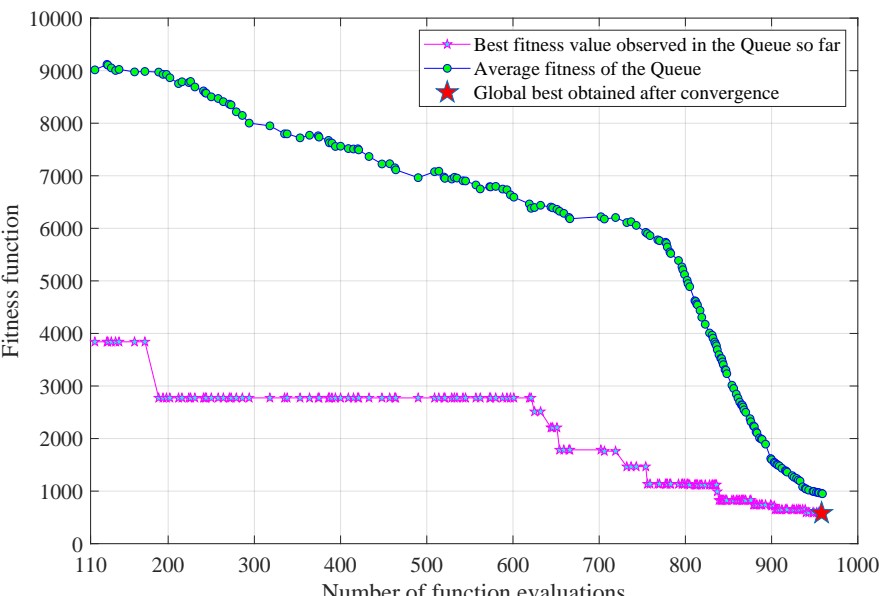

**Figure 5.** Convergence curve for the modified CE algorithm implemented to optimize the metallic patch dimension of the PGM. The plot compares the evolution of the average fitness of the queue with respect to the best fitness of the queue so far.

The average fitness gradually converges toward the vicinity of the best fitness, signifying the convergence of the optimization process. Notably, the curve's central segment indicates that the degree of smoothing applied is quite pronounced, and there might be potential for reducing it to accelerate convergence. Past experience with the CE method anticipates that a higher value of $\alpha$ would likely hasten the convergence rate. Nonetheless, it is important to acknowledge the inherent trade-off between computational expenditure and precision. The algorithm was stopped once the diversity among the elite candidates dropped below the predefined threshold.

### 4.2. Optimizing Amplitude Distribution in the Feed Array of a PGM to Control SLLs

In various satellite applications, metasurfaces designed for beam steering are often combined with high-gain feeding apertures, such as arrays of closely arranged low-gain antennas. These arrays create a narrow, highly focused beam directed perpendicularly (broadside) to the array plane. They achieve this with a near-uniform phase distribution across the near-electric field. However, when integrating a beam-steering metasurface that introduces a gradual phase progression within the near-electric field of the feeding array, the previously perpendicular beam of the array undergoes a tilt at an off angle. Unfortunately, this can lead to the emergence of multiple undesired significant sidelobes in the far-field radiation pattern.

Analogies can be drawn between the theory of antenna array pattern synthesis and the control of metasurface beam steering. In this context, the methodology for designing metasurfaces can be likened to the discrete sampling process employed in antenna arrays. According to antenna array theory, effective control over the sidelobe level (SLL) can be achieved by utilizing appropriate amplitude distributions in the excitation field [36]. In the realm of phased arrays, techniques such as Chebyshev or Taylor amplitude distributions have been applied to regulate SLLs within the far-field pattern [37–39]. Building upon this concept, we posit that an optimized amplitude excitation has the potential to efficiently suppress undesired sidelobes within the far-field pattern of antennas based on metasurfaces.

To explore this concept, we use the design configuration shown in Figure 6, which offers a side view of the setup used for this investigation.

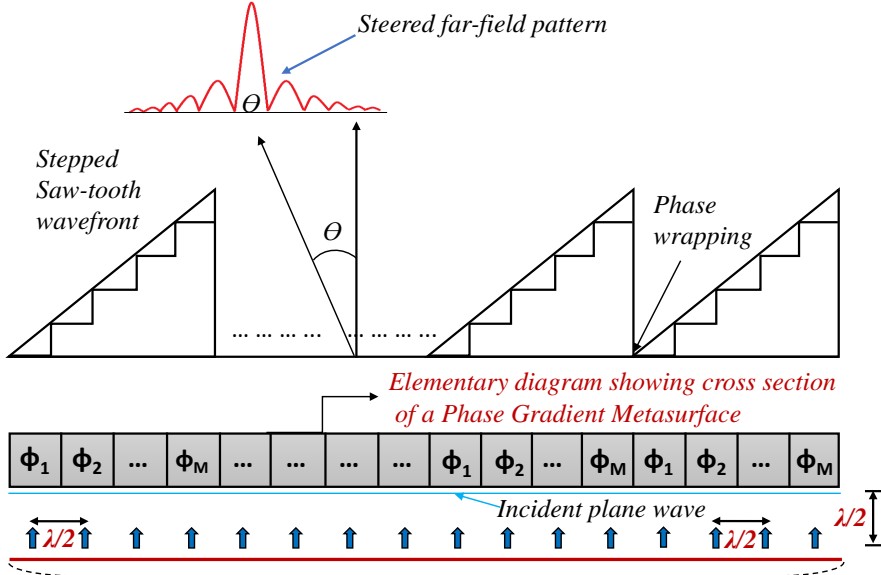

**Figure 6.** Elementary diagram for the side view of an array-fed beam-steering PGM, showing a stepped saw-tooth phase profile in the near field and a steered pattern in the far field.

The illustration depicts an array feed situated at a half-wavelength distance beneath the metasurface, giving rise to a plane-wave propagation along the *z*-axis. This feed encompasses an array of infinitesimal Hertzian dipoles spaced half a wavelength apart, all positioned in front of a ground plane. A uniform magnitude and phase are assigned to the feed's elements. Each spatial phase-shifting cell within the metasurface introduces a specific phase modification, thereby establishing a consistent progressive phase disparity between adjacent cells. Consequently, the metasurface output assumes a continuous, linearly escalating phase profile. This phase profile, upon cycling every 360° increment in the phase delay, transforms into a saw-tooth phase distribution. The unchanging phase

difference between neighboring cells ($\Delta\phi = {}^{2\pi}/M$) remains constant, where M denotes the count of unique cells in the supercell (one period within a PGM is denoted as a supercell).

The distinct phase delays ($\phi_1, \phi_2 \dots \phi_M$) undergo periodic repetition to align with the recurring structure of a supercell within the metasurface. This linearly augmenting phase, equivalent to an unwrapped saw-tooth phase, induces beam tilting ($\theta$) within the far-field radiation pattern. The metasurface with an increasing electric field phase variation, as shown in Figure 6, yields a steered beam but with sidelobes as well as periodic grating lobes. Ideally, the wavefront of a steered beam should have a periodically repeated saw-tooth profile [40]. However, owing to the inherent phase discretization integral to the metasurface design, a stepped saw-tooth wavefront is obtained in reality. Real-world conditions introduce factors like mutual coupling, fringing fields, assumptions of periodicity in design, abrupt geometric alterations, and manufacturing imperfections, all of which collectively contribute to pattern deterioration. Consequently, deviations from the anticipated saw-tooth phase profile occur. The outcome is an elevation in the sidelobe levels (SLLs) within the far-field radiation pattern. The approach to metasurface design can be analogously likened to the discrete sampling method employed in aperture-based antenna arrays.

The generalized CE algorithm is implemented to optimize the amplitude distribution within the feed array of a PGM. The primary objective is to mitigate the presence of undesirable sidelobe levels (SLLs) within the far-field radiation pattern. This study delves into leveraging the amplitude variation within the feed array to proficiently govern the extent of undesired lobes in a beam-steering antenna based on a PGM. While the manipulation of the phase through the metasurface facilitates main-lobe steering and control across the complete 0 to $2\pi$ range, the optimization of the excitation amplitudes within the feed array provides a strategic avenue for effective SLL management.

### 4.2.1. Problem Formulation

Antenna array theory has established that sidelobe levels (SLLs) can be controlled through customized amplitude distributions within the excitation field [36]. In our methodology, we harness this principle by supplying each element of the supercell with infinitesimally small Hertzian dipoles. Notably, distinct excitation amplitudes are assigned to each dipole element while maintaining a constant phase, as illustrated in Figure 7.

The design variables $V_1, V_2 \dots V_N$ represent the set of feed amplitudes, which are recurrently replicated. Our objective revolves around identifying an optimal arrangement of excitation amplitudes capable of effectively mitigating SLLs to levels below the constraints stipulated in the FCC mask outlined in (10). For the purpose of optimization, the simplified 1D metasurface model depicted earlier in Figure 3 is utilized. The parameter vector consists of a periodic sequence of amplitude distributions incorporating $N$ distinct amplitude values.

$$\mathbf{x} = (V_1, V_2, V_3, V_4, \dots, V_N), \tag{12}$$

where $(V_1, V_2, V_3, V_4, \dots, V_N) \in [0, 1]$. The fixed parameters of a Hertzian dipole array-fed metasurface design are the aperture size ($23\lambda_0 \times 23\lambda_0$), cell size ($d = {}^{\lambda_0}/2$), and permittivity of the dielectric substrate ($\varepsilon_r = 2.2$), along with the patch and hole dimensions (a1, a2, a3, a4, a5, b1, b2, b3, b4, b5, r), whose values are tabulated in the next section.

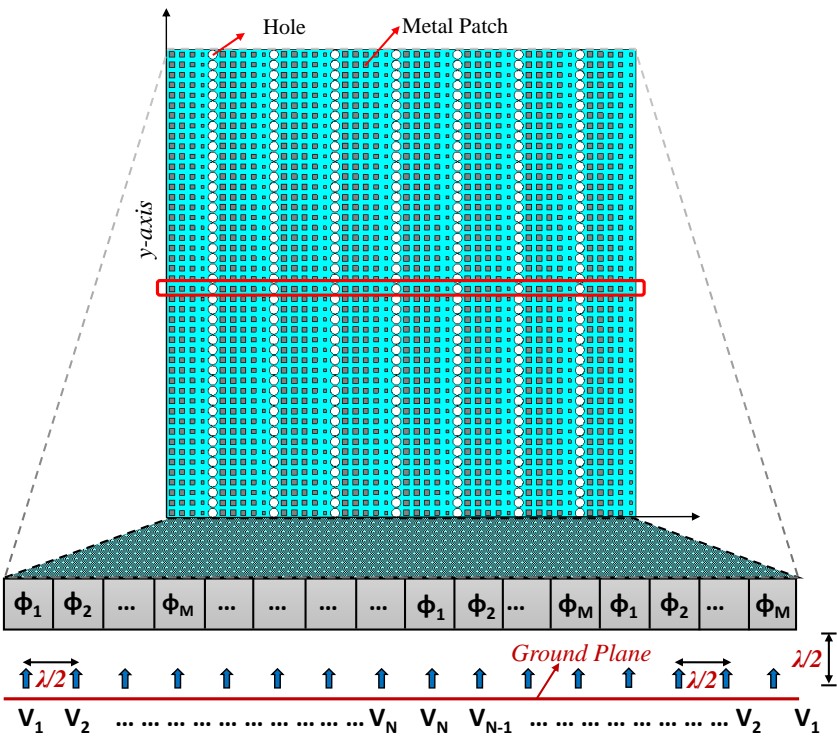

**Figure 7.** Elementary diagram of the side view of an array-fed beam-steering PGM, showing a stepped saw-tooth phase profile in the near field and a steered pattern in the far field.

### 4.2.2. Algorithm Implementation

The process of optimizing the distribution of feed amplitudes constitutes a continuous optimization problem confined within a specific search space. By attributing distinct amplitude excitations to each element within the supercell, effective control over the sidelobe levels within the radiation pattern is achieved. The optimization parameters $(V_1, V_2 \ldots V_N)$ are permitted to vary continuously within the interval of 0 to 1. To maintain adherence to this range, population candidates are sampled independently and identically (i.i.d) from a beta probability distribution function $(f(x|\alpha, \beta))$, as shown in Figure 8, since it naturally supports the bound within $[0, 1]$. The initial distribution uses $\alpha = 1$ and $\beta = 1$.

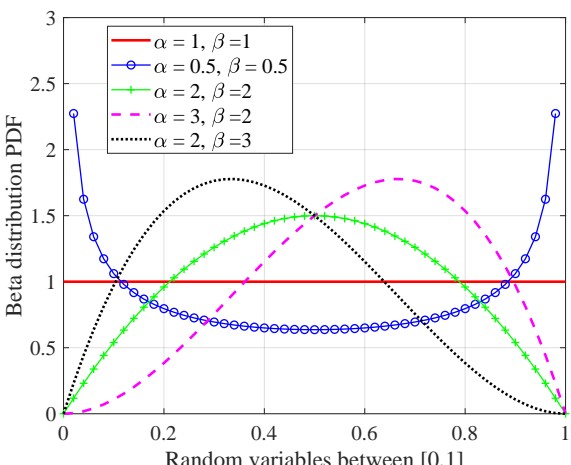

**Figure 8.** Beta distribution probability density functions for different sets of values of parameters $\alpha$ and $\beta$.

After each evaluation, the probability distribution parameters are updated using the elite population, as detailed in Algorithm 1. The fitness function (*FF*), detailed in (11),

is used with the objective of ensuring that the directivity pattern aligns with the FCC mask (25.209) for the Ka-band, as defined in (10). To facilitate precise full-wave EM simulation-driven optimization, the generalized CE algorithm is implemented in MATLAB and seamlessly integrated with CST MWS. This iterative algorithm progressively minimizes the fitness function, consequently diminishing undesired sidelobe levels (SLLs) within the radiation pattern. The Chebyshev or Taylor amplitude tapering exhibits a symmetric distribution characterized by a bell-shaped profile. To replicate a similar pattern, we incorporate 23 distinct amplitude excitations. These excitations are duplicated in a mirrored symmetry to encompass the entire aperture, consisting of 46 elements along the x-direction. This arrangement is illustrated in Figure 9.

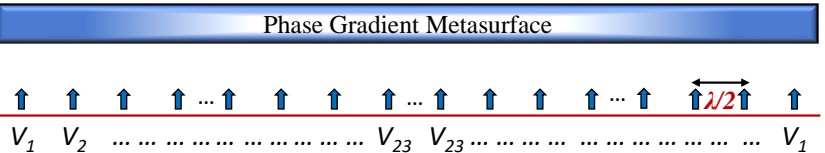

*Hertzian dipole array periodically distributed with period N=23*

**Figure 9.** A PGM fed with an array of Hertzian dipoles with an amplitude excitation periodicity of $N = 23$.

Table 2 lists the user-defined input parameters set in the algorithm to optimize the feed amplitude distributions.

**Table 2.** The CE algorithm parameters for the optimization of excitation amplitude distributions with a periodicity of $N = 23$ in the feed array of a PGM.

| Queue Size $N_q$ | Elite Sub-Population Size $N_{el}$ | Smoothing $\alpha_S$ |
|:---:|:---:|:---:|
| 150 | 15 | 0.5 |

### 4.2.3. Optimization Results

On an Intel Core i7-6700 CPU running at a clock speed of 3.4 GHz and equipped with 64 GB of RAM, each evaluation carried out within the CST MWS time-domain solver required approximately 21 min. This was notably faster than the previous approach, as the mesh remained unchanged during each evaluation due to the fixed metallic patch dimensions. The entire optimization process involving 23 design variables spanned a cumulative duration of 447 h and 48 min, conducting a total of 566 function evaluations before the termination criterion was met. It is worth highlighting that in both optimization cases, the computation time was extended due to the lack of GPU acceleration, which could have significantly expedited the optimization process. The initial 150 function evaluations were dedicated to queue initialization. This step involved populating the queue with candidates and their corresponding fitness values. Following this initial phase, subsequent evaluations contributed to identifying the elite candidates and subsequently updating the probability distribution parameters. This dynamic process directed the search toward the global optimum. The convergence results, as depicted in Figure 10, showcase the evolution of both the average fitness across all samples within the queue and the best fitness observed up until that point. The graph demonstrates that the optimal solution was attained at the 508th function evaluation.

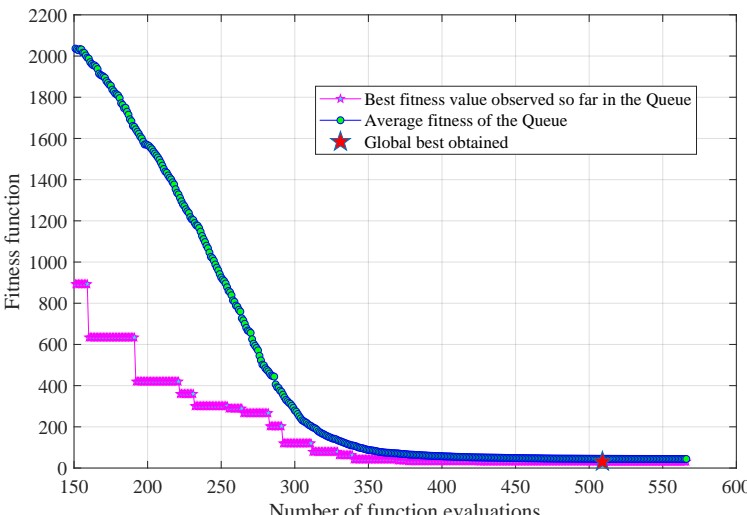

**Figure 10.** Convergence results using the modified CE method for Case-II optimization of the PGM.

## 5. Electromagnetic Simulation Results

In the context of the first optimization strategy, the design characteristics of the PGM before and after optimizing the dimensions of the metallic patches are presented in Table 3.

**Table 3.** Design parameter values for the PGM before and after optimization.

| Design Parameters, mm | Before Optimization | After Optimization |
|---|---|---|
| $a_1, b_1$ | 2.27, 2.25 | 2.54, 2.09 |
| $a_2, b_2$ | 3.30, 3.50 | 3.75, 3.47 |
| $a_3, b_3$ | 3.88, 3.85 | 3.65, 4.12 |
| $a_4, b_4$ | 4.11, 4.14 | 4.28, 3.97 |
| $a_5, b_5$ | 4.30, 4.20 | 4.10, 4.44 |
| $r$ | 3.4 | 3.68 |
| Sidelobe Level, dB | $-13$ | $-16.5$ |
| FF, Equation (11) | 4284 | 561 |

The optimized design parameters were employed to construct a finite-sized 2D metasurface structure. This metasurface was subsequently subjected to simulation using the CST MWS time-domain solver, with excitation through a waveguide port and employing "Open-Add space" boundary conditions in both the x- and y-directions.

In Figure 11, we can observe and compare the directivity patterns obtained through full-wave electromagnetic simulation for both the initial and optimized finite-sized 2D metasurfaces. In the initial design, the directivity pattern of the PGM exhibited deviations from the FCC mask at four distinct points ($\theta = -80°, -20°, 0°, 40°$) within the observable range of $\theta$. After undergoing the optimization process, a substantial portion of the sidelobes was effectively mitigated, and the directivity pattern violated the FCC mask at only two locations: $\theta = -40°$ by 3.5 dB and $\theta = 40°$ by 0.5 dB. Furthermore, the SLL was reduced by 4 dB (from $-13$ dB to $-17$ dB) compared to the initial design.

The results obtained using this approach represent solutions that are in close proximity to the global optimum while simultaneously preserving the structural integrity and not disrupting the underlying physics. Moreover, this method acknowledges the delicate balance between achieving the desired results and the time invested in the optimization process. Consequently, the stopping criterion was not rigidly set to FF = 0, where the fitness value reaches its absolute minimum. The definition of the stopping criterion is a critical factor influencing the algorithm's convergence. In this particular case, the optimizer concludes its operation when the diversity among the sampled candidates in the queue drops below a predefined threshold. At this juncture, the best parameters discovered are

regarded as the global optimum. It is worth noting that this global optimum may differ slightly from the true global best, but it remains effective in addressing the specific design problem under consideration.

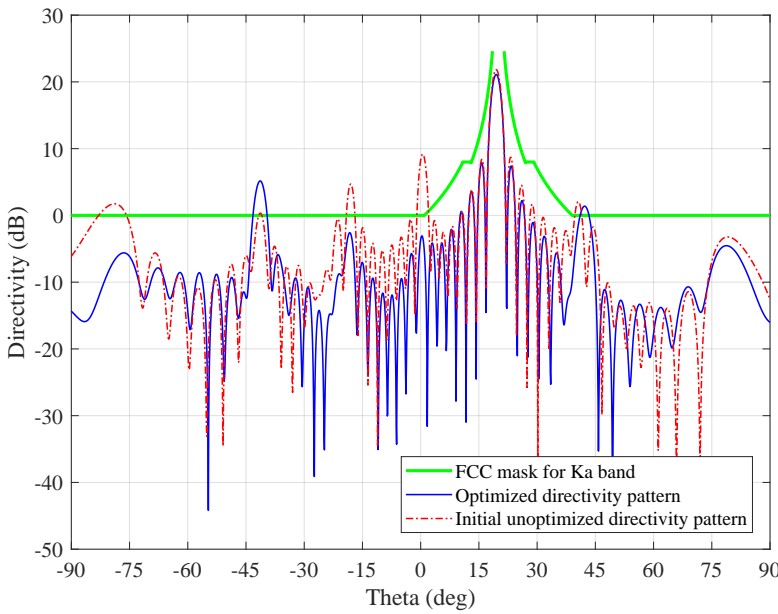

**Figure 11.** Directivity pattern comparison of optimized and unoptimized PGMs.

Likewise, in the context of the second optimization strategy, Table 4 presents the excitation amplitude parameters of the PGM both before and after the optimization process. In CST MWS, a finite-sized 2D metasurface was subject to simulation, employing "Open-Add space" boundary conditions in both the x- and y-directions. The metasurface was then excited using a 2D array of Hertzian dipoles featuring the optimized excitation amplitude distribution. Subsequently, the results were compared with those obtained from a simulation where a 2D array of Hertzian dipoles was uniformly excited as the source.

**Table 4.** Excitation amplitude distribution of feed array before and after optimization.

| Design Parameters, mm | Before Optimization | After Optimization |
|---|---|---|
| $V_1$ | 1 | 0.34 |
| $V_2$ | 1 | 0.70 |
| $V_3$ | 1 | 0.65 |
| $V_4$ | 1 | 0.55 |
| $V_5$ | 1 | 0.74 |
| $V_6$ | 1 | 0.34 |
| $V_7$ | 1 | 0.70 |
| $V_8$ | 1 | 0.65 |
| $V_9$ | 1 | 0.55 |
| $V_{10}$ | 1 | 0.74 |
| $V_{11}$ | 1 | 0.34 |
| $V_{12}$ | 1 | 0.70 |
| $V_{13}$ | 1 | 0.65 |
| $V_{14}$ | 1 | 0.55 |
| $V_{15}$ | 1 | 0.74 |
| $V_{16}$ | 1 | 0.74 |
| $V_{17}$ | 1 | 0.34 |
| $V_{18}$ | 1 | 0.70 |
| $V_{19}$ | 1 | 0.65 |
| $V_{20}$ | 1 | 0.55 |
| $V_{21}$ | 1 | 0.74 |
| $V_{22}$ | 1 | 0.34 |
| $V_{23}$ | 1 | 0.46 |
| Directivity, dBi | 21.9 | 20.5 |
| Sidelobe Level, dB | −12.4 | −19.3 |
| FF, Equation (11) | 1247 | 32.8 |

Figure 12 illustrates a comparison of the directivity patterns of the PGM before and after the optimization of the excitation amplitude distribution. Notably, the optimization endeavor successfully suppressed the undesirable sidelobes, effectively keeping them below the FCC mask, with the exception of a single sidelobe situated at $-20°$. Additionally, the directivity of the optimized PGM experienced a reduction of 1.4 dB. This reduction can be primarily attributed to the tapering introduced within the feed array, which played a pivotal role in achieving these improvements.

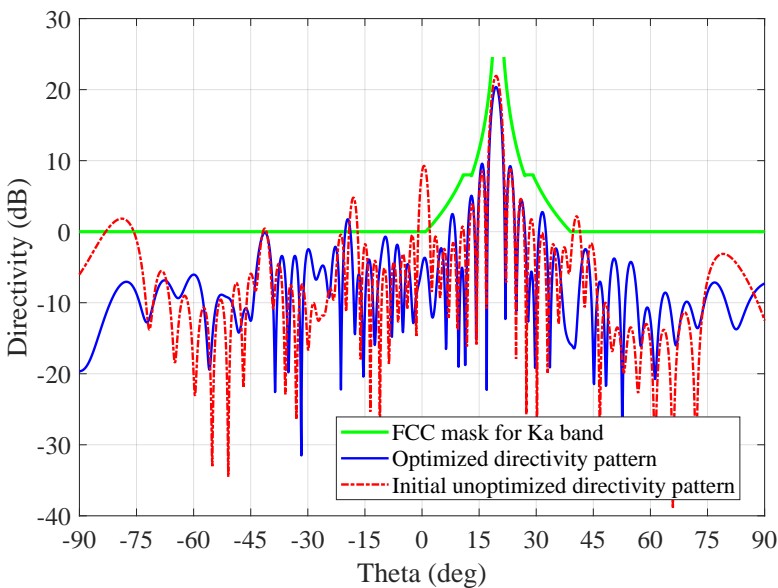

**Figure 12.** Comparison of the radiation pattern of a finite-sized 2D metasurface when excited by a 2D array of Hertzian dipoles, showing a decrease in sidelobe levels (SLLs) following the optimization of the excitation amplitude distribution compared to a uniformly excited array.

The overall directivity decreased from 21.9 dBi to 20.5 dBi (1.4 dB reduction) due to the tapering in the amplitude distribution achieved after optimization. The SLL was reduced by 6.9 dB (from $-12.4$ dB to $-19.3$ dB).

## 6. Discussion

Managing sidelobe levels (SLLs) in metasurface-driven beam-steering antennas presents a formidable challenge due to the involvement of various intrinsic factors that contribute to the emergence of grating lobes. The inherent complexity of metasurface structures adds to the difficulty in achieving precise control over these sidelobes. In response to this challenge, we propose a method that leverages an equivalent model, rendering the optimization of electrically large periodic metasurfaces more feasible and computationally efficient. This model is designed to provide accurate predictions of the complete metasurface performance. It incorporates an antenna array factor calculator that duly accounts for mutual coupling between the metallic patches. By optimizing the PGM-based beam-steering antenna and strategically mitigating excessive sidelobes, we introduced a streamlined evolutionary optimization algorithm rooted in the CE method. This innovative approach effectively addresses the intricacies of sidelobe suppression, contributing to the attainment of enhanced far-field radiation pattern performance.

To achieve a beam-steered radiation pattern that complies with the FCC mask (25.209) for Ka-band applications and effectively suppresses undesired sidelobes, we adopted two distinct strategies. In the first approach, we optimized the dimensions of the patches, aiming to attain a radiation pattern devoid of spurious sidelobes. This optimization effort resulted in a notable reduction of the sidelobe levels (SLLs), thus significantly enhancing the overall metasurface performance. The second approach maintained the dimensions of the PGM constant while focusing on optimizing the amplitudes within the dipole array

feed. This amplitude optimization of the feed array yielded outcomes similar to those achieved through the first approach, particularly in terms of sidelobe management.

The outcomes from these optimization cases offer valuable insights into the effectiveness and applicability of the cross-entropy method for CPU-intensive electromagnetic optimization challenges. Additionally, they provide essential guidance for informed decision-making when considering this optimization approach for analogous applications. A significant drawback of conventional population-based optimization methods lies in their heavy reliance on an extensive number of forward solver calls. This reliance renders them impractical when dealing with computationally demanding full-wave time-domain electromagnetic (EM) simulations, particularly in cases involving intricate EM structures that necessitate substantial computational resources and time.

In response to this challenge, the proposed variant of the cross-entropy (CE) method addresses this limitation by retaining the simplicity and elegance of the CE approach while simultaneously enhancing monitoring capabilities. This improvement facilitates the observation of the algorithm's progress. Notably, it has been observed that employing smaller generations leads to more favorable expected improvements per function evaluation. This adapted version of the CE method not only demonstrates its efficiency and potential in the design and optimization of metasurfaces but also showcases its versatility across various complex EM design problems. As a result, it presents an appealing alternative to other commonly used optimization algorithms. With its favorable convergence properties, the proposed CE method establishes itself as a competitive and efficient solution for addressing computationally intensive EM optimization challenges.

In contrast to the previously proposed PGM optimization strategies documented in Singh et al.'s works [1,2], which rely on a simplified supercell model and exclusively exploit surface periodicity, the method presented here employs a more precise equivalent model to anticipate the performance of finite-sized metasurfaces. The proposed optimization approach relies solely on the radiation pattern of the 1D metasurface, forecasted after conducting full-wave electromagnetic simulations, treating the problem as a black-box optimization. This distinguishes it from the alternative methods outlined in Singh et al.'s publications [1,2], which delve into the intricacies of the simulation model's physics.

The outcomes projected by our proposed optimization approach closely align with real-world performance since the equivalent model applies the periodicity assumption exclusively along the y-direction while maintaining the x-direction boundary conditions identical to the original metasurface. In the case of significantly large PGM apertures, Floquet analysis-based optimization approaches using supercells can significantly reduce computation time. However, they do not accurately predict the far-field pattern of finite-sized PGMs, where supercell repetitions are fewer, or truncated metasurfaces. For metasurfaces with fewer supercell repetitions, the optimization approach proposed in this work offers more reliable predictions and accurate optimization results.

**Author Contributions:** Concept K.S.; Methodology, K.S.; Software, K.S.; Validation, K.S.; Formal analysis, K.S.; Investigation, K.S.; Resources, K.E.; Writing—original draft, K.S.; Writing—review & editing, K.S. and K.E.; Visualization, K.E.; Supervision, K.E.; Funding acquisition, K.E. All authors have read and agreed to the published version of the manuscript.

**Funding:** iRTP Scholarship given to Khushboo Singh by Macquarie University.

**Data Availability Statement:** Not applicable.

**Conflicts of Interest:** The authors declare no conflict of interest.

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
