# Peer review of "Suppressing Sidelobes in Metasurface-Based Antennas Using a Cross-Entropy Method Variant and Full Wave Electromagnetic Simulations"

_electronics, doi:10.3390/electronics12204229_

Round 1

Reviewer 1 Report

In this paper, the authors present a streamlined evolutionary optimization algorithm rooted in the CE method to solve the challenges of sidelobe suppression. By appropriately optimizing the dimensions of patches and the amplitude of constant PGM, the undesired sidelobes are effectively suppressed and the far-field radiation pattern still maintains excellent performance. However, there are some important issues that need improvement, and the specific comments are as follows:

Comment 1: Please justify the novelty of the paper compared to the same works. The author could consider adding the information in the abstract and at the end of the introduction. Besides, some more detailed elaboration is needed to demonstrate the practicality of this innovation.

Comment 2: The authors have simulated the performance of the proposed structure, but the electromagnetic properties have not been validated by the experiments. Please provide the measurement result of the corresponding sample.

Comment 3: Please specify the hardware parameters that the proposed optimization method requires, and add the time consumption of the whole operation.

Comment 4: (minor) Do you consider the coupling effect between the adjacent unit cells when adopting the optimization algorithm to obtain the ideal array periodical distribution?

There are still some minor errors, please check the whole manuscript carefully.

Author Response

Reviewer: 1

Recommendation:

In this paper, the authors present a streamlined evolutionary optimization algorithm rooted in the CE method to solve the challenges of sidelobe suppression. By appropriately optimizing the dimensions of patches and the amplitude of constant PGM, the undesired sidelobes are effectively suppressed and the far-field radiation pattern still maintains excellent performance. However, there are some important issues that need improvement, and the specific comments are as follows:

Comment # 1:

Please justify the novelty of the paper compared to the same works. The author could consider adding the information in the abstract and at the end of the introduction. Besides, some more detailed elaboration is needed to demonstrate the practicality of this innovation.

Authors’ Response:

Thank you for highlighting this aspect, as the article did not explicitly discuss the novelty of the manuscript and practicality of the method was not well explained and was not clear enough for readers. In contrast to previously presented related articles, this manuscript offers an intricate exposition of the optimization process, along with a rationale for choosing specific probability distribution functions over others. The detailed elucidation of the cross-entropy method variant contributes a robust conceptual framework that can be applied to numerous intricate problems. Notably, no prior work has delved into the comprehensive analysis of this particular CE-method variant, making it a pivotal element. Furthermore, the optimization results presented in this study diverge from those in prior publications. In contrast to the periodic feed distribution discussed previously in [9], our approach assumes radial symmetry, which aligns more closely with the ideal amplitude scenario for feed tapering, such as Taylor taper or Chebyshev taper distribution.

When the metasurface was rotated over a periodic amplitude-distributed feed array, the sidelobes tend to increase. Thus, we used a radially symmetric feed distribution that does not affect the sidelobes when the metasurface is rotated over the feed array to steer the beam in azimuth plane. (This analysis is a part of a different study that authors are conducting and wish to present in future work.)

In this work we also compare the two different strategies which can be implemented individually or combined together to achieve enhanced performance with reduced sidelobes and grating lobes in a metasurface-based beam-steering antenna system.

To justify the novelty of the work, following additions have been made in the abstract of the revised manuscript.

“The uniqueness of the proposed optimization strategy lies in its utilization of an equivalent 1D metasurface model for optimization that not only considers the mutual coupling between identical unit cells along the y-direction within a complete metasurface but also takes into account the distinct cells along the x-direction. Moreover, the 1D metasurface model also incorporates the influence of edge effects along the x-direction.”

To address the concerns regarding similarity with work already presented by the authors in a conference article, following additions have been made in the introduction of the revised manuscript. Note: Figure 7 has also been updated accordingly.

“In contrast to the previously presented related article [8], which is essentially a very concise study, this manuscript provides an intricate exploration of the optimization process, along with a justification for selecting specific probability distribution functions over alternatives. The detailed clarification of the cross-entropy method variant contributes a sturdy conceptual framework that can be applied to a variety of complex problems. Notably, there has been no prior comprehensive analysis of this particular variant of the CE method in the context of electromagnetic optimization, making it a pivotal component. Furthermore, it's important to note that the optimization results presented in [8] were based on a periodic feed distribution, in contrast to our approach where we employ a radial symmetry in the feed amplitude distribution. This radial symmetry aligns more closely with the ideal scenario for feed tapering, such as the Taylor or Chebyshev taper distributions. In this research, we also conduct a comparative analysis of two distinct optimization strategies that can be implemented either separately or in combination. The objective of this analysis is to enhance overall performance by reducing sidelobes and grating lobes in a metasurface-based beam-steering antenna system.

Comments have been added regarding the practicality of the proposed approach in the discussion section on page of the revised manuscript.

“The outcomes projected by our proposed optimization approach closely align with real-world performance since the equivalent model applies the periodicity assumption exclusively along the y-direction while maintaining the x-direction boundary conditions identical to the original metasurface. In the case of significantly large PGM apertures, the Floquet analysis-based optimization approaches using supercells can significantly reduce computation time. However, they do not accurately predict the far-field pattern of finite-sized PGMs where supercell repetitions are less. For PGMs with lesser supercell repetitions, the optimization approach proposed in this work offers more reliable predictions and accurate optimization results.”

Comment # 2:

The authors have simulated the performance of the proposed structure, but the electromagnetic properties have not been validated by the experiments. Please provide the measurement result of the corresponding sample.

Authors’ Response:

We understand that experimental validations are essential to validate most of the full-wave electromagnetic simulation results. However, we want to emphasize that the primary objective of this article is to underscore the significance of optimization strategies. Given that metasurfaces constitute electrically large structures, the conventional simulation-based optimization process presents significant challenges. In this approach, we harness the benefits of evolutionary algorithms to optimize the structure while keeping computational costs at a minimum. We have validated our approach through simulations using CST. Of course, this manuscript does not include experimental results. It's important to note that our ultimate goal is to integrate these optimized metasurfaces as add-on devices with high-gain base antennas in future applications such as beam-offset and beam scanning. Providing the experimental results requires the integration of these surfaces with an appropriate base antenna needs further study and intensive analysis. However, the results pertaining to this broader investigation are outside the scope of this specific article. We believe that all the missing important information has been added in the revision.

Comment # 3:

Please specify the hardware parameters that the proposed optimization method requires, and add the time consumption of the whole operation.

Authors’ Response:

The hardware parameters are provided in the manuscript for the patch-dimension optimization in the “Optimization Results”- sub-subsection on page 10, as quoted below:

“Each evaluation conducted within the CST MWS time-domain solver necessitated approximately $28$ minutes, utilizing an Intel Core i7-6700 CPU clocked at $3.4$~GHz, accompanied by a $64$~GB RAM. The entirety of the optimization procedure, which encompassed 959 function evaluations, spanned a cumulative duration of 447 hours and 48 minutes.”

For second approach of feed-amplitude distribution optimization, we created a new sub-subsection “Optimization Results” and added the following text in the revised manuscript on page 14:

“Optimization Results

On an Intel Core i7-6700 CPU running at a clock speed of 3.4 GHz, and equipped with 64 GB of RAM, each evaluation carried out within the CST MWS time-domain solver required approximately 21 minutes. This is notably faster than the previous approach, as the mesh remained unchanged during each evaluation, thanks to the fixed metallic patch dimensions. The entire optimization process involving 23 design variables spanned a cumulative duration of 447 hours and 48 minutes conducting a total of 566 function evaluations before the termination criterion was met. It is worth highlighting that in both optimization cases the computation time was extended due to the lack of GPU acceleration, which could have significantly expedited the optimization process.”

Comment # 4:

Do you consider the coupling effect between the adjacent unit cells when adopting the optimization algorithm to obtain the ideal array periodical distribution?

Authors’ Response:

We extend our appreciation to the reviewers for highlighting this point, as many other electromagnetic optimizations based on analytical approaches often rely on several assumptions. In contrast to those, the optimization methods employed here take into consideration the coupling effect between the adjacent non-identical unit cells along x-direction as well the identical unit cells along y-direction. The 1D metasurface model selected for optimization purpose considered all the unit-cells present in the actual finite sized metasurface along x-direction since they only exhibit supercell periodicity along this direction and are essentially locally non-periodic. However, since the metasurfaces are periodic at the unit-cell level along y-direction, the mutual coupling effect can be modelled by using period boundary conditions, as the actual finite sized metasurface basically repeats the unit cells sufficient number of times to mimic infinite periodicity.

Another striking feature of this approach is the fact that we also include the edge effects along the x-direction, unlike the other presented optimization approach in [2].

We have included the details regarding mutual coupling and edge-effects in the revised manuscript on page 7. The exact text is as follows:

“Unlike many other electromagnetic optimizations based on analytical approaches, which often rely on various assumptions, our optimization methods stand out by taking into account the coupling effect between adjacent non-identical unit cells along the x-direction, as well as the identical unit cells along the y-direction in this simplified equivalent model. In our optimization of the 1D metasurface model, we consider all the unit cells present in the actual finite-sized metasurface along the x-direction. This is because they only exhibit supercell periodicity along this direction and are essentially locally non-periodic. However, along the y-direction, metasurfaces are periodic at the unit-cell level, so we model the mutual coupling effect using periodic boundary conditions. The actual finite-sized metasurface effectively repeats the unit cells a sufficient number of times to mimic infinite periodicity in this direction. Another noteworthy feature of our approach is the inclusion of edge effects along the x-direction. This is a departure from other optimization approaches presented in [1, 9], where such edge effects were not considered.”

Comment # 5:

Comments on the Quality of English Language. There are still some minor errors, please check the whole manuscript carefully.

Authors’ Response:

We have thoroughly reviewed the manuscript and also used Grammarly to make necessary changes and corrections to rectify any errors. To the best of our abilities, we believe that the manuscript is now free of errors. The changes made in the manuscript are highlighted in red color over the entire manuscript.

Reviewer 2 Report

Is the proposed method is more productive (faster) than the existing ones? Give some comparison in terms of time or capabilities with the optimizer from CST (as example).

It is not entirely clear how the optimization problem was formulated in Eq.8. Is an integer number of supercells along PGM set as additional condition? 

In accordance with data presented in Fig.12 the goal of optimization has not been achieved due to a sidelobe at -20 degrees. The authors believe that this is the best possible result for the selected elements' geometry? Or was there no further optimization carried out due to time constraints?

The method proposed in this article (with the same simulation results) has already been presented by the authors earlier in DOI: 10.23919/URSI-EMTS.2019.8931531

Author Response

Reviewer: 2

Comment # 1:

Is the proposed method is more productive (faster) than the existing ones? Give some comparison in terms of time or capabilities with the optimizer from CST (as example).

Authors’ Response:

We thank the reviewer for the time and insightful comments that has greatly helped to improve the manuscript. The world of global evolutionary optimization methods is extensive, involving many techniques. Among the most prominent ones are Genetic Algorithms, Particle Swarm Optimization (PSO), and Covariance Matrix Adaptation Evolution Strategies (CMA-ES), which have been widely employed in addressing complex optimization challenges in recent years. Having said that, is indeed essential to establish and justify the potential of the cross-entropy method to perform on par with or even surpass the performance of these well-established methods. In this context, we would like to bring the reviewer's attention to the work conducted by one of our team members (DOI: 10.1109/JMMCT.2020.3000563), which specifically compared the cross-entropy method with PSO and CMA-ES. The findings revealed that while all three methods ultimately reached a similar solution, the cross-entropy method exhibited a notable advantage in terms of speed. This empirical evidence strongly supports the assertion that the cross-entropy method is a faster option for electromagnetic optimizations.

We have added following text to the revised manuscript in the introduction section on page 2 of the revised manuscript.

“The domain of global evolutionary optimization methods encompasses a diverse array of techniques. Notably, Genetic Algorithms (GAs), Particle Swarm Optimization (PSO), and Covariance Matrix Adaptation Evolution Strategies (CMA-ES) emerge as prominent contenders [14]. These methodologies have gained widespread traction in addressing intricate optimization challenges within the field of electromagnetics in recent years.

It is imperative not only to establish but also to substantiate the efficacy of the cross-entropy method in achieving performance parity with or possibly surpassing these firmly established techniques. In order to validate the capabilities of the cross-entropy method, we draw the reader's attention to the investigations made in [14]. This study specifically undertook a comparative assessment of the cross-entropy method against PSO and CMA-ES. The findings elucidated that while all three methods ultimately converged towards similar solutions, the cross-entropy method demonstrated a pronounced advantage in terms of computational efficiency. This empirical evidence strongly supports the assertion that the cross-entropy method is a faster option for electromagnetic optimizations.”

CST Microwave Studio (CST MWS) offers both built-in local optimization techniques and some global optimization methods like Genetic Algorithms (GA), Particle Swarm Optimization (PSO), and Covariance Matrix Adaptation Evolution Strategies (CMA-ES). However, the local optimization algorithms within CST MWS often fall short in optimizing complex structures to achieve optimal solutions.

While the global optimizers in CST MWS serve as valuable tools for attaining a certain level of optimization, they do have notable limitations. Specifically, users are often constrained by their inability to modify most of the internal optimization parameters. Additionally, defining a fitness function can be a challenging task, as it may not accommodate all types of fitness function equations.

Another limitation lies in the lack of accessibility to intermediate optimization results, leaving users largely unaware of the progression of the optimization process. CST MWS provides limited user control over the optimization process.

In contrast, interfacing a MATLAB code with CST MWS offers greater flexibility for post-processing, variable definition, and sampling choices. This extended control allows users to overcome some of the limitations inherent in the optimization capabilities of the in-built algorithms in CST MWS optimizer.

Comment # 2:

It is not entirely clear how the optimization problem was formulated in Eq.8. Is an integer number of supercells along PGM set as additional condition? 

Authors’ Response:

We appreciate the reviewer for bringing up this crucial point. Upon careful review, the authors have acknowledged that this section was not presented clearly, and there were numerical inaccuracies. The 1D equivalent model repeats the supercell along the x-direction, which can be seen in Fig.4. Thus, essentially the design variables also follow the same repetitions and the variables are defined considering this repetitive pattern. Also, the upper and lower limits have been changed after cross-checking from the code. The lower limit is 0.05 mm for both patch dimension and hole radius, while the upper limit is 7 mm for patch length and 3.5 mm for hole radius. These, limits ensure that the model is feasible and there is no over lap between the holes or the metallic patches, since each unit element is 7.5 mm.

Reviewer's question has raised authors' interest in exploring the optimization process with distinct dimensions for each design parameter. The authors are considering conducting this investigation in the near future. We believe that the proposed cross-entropy method can efficiently handle large number of design variables.

To rectify these issues, we have made revisions to the manuscript on page 9, and the specific changes implemented are outlined below: The text highlighted in red are the actual changes made to the manuscript. The blue text has been kept from the previous version.

“As the supercells are essentially replicated along the x-direction, the design dimensions of the metallic strips and holes within the supercell also follow this repetition across the metasurface. Consequently, the side lengths of the first (top) and third layer metal patches (a1, a2, a3, a4, a5), as well as the
second and fourth (bottom) layer metal patches (b1, b2, b3, b4, b5) for five consecutive cells,
along with the radius (r) of the hole within the supercell (as depicted in Fig. 4), collectively
form an 11-parameter design vector, as defined in Equation (7):

x = (a1, a2, a3, a4, a5, b1, b2, b3, b4, b5, r). (7)

The optimization problem for the metasurface design is formulated with a set of inequality
constraints, as represented in Equation (8).

{0.05 mm ≤ a1, a2, a3, a4, a5, b1, b2, b3, b4, b5 ≤ 7 mm

0.05 mm ≤ (r) ≤ 3.5 mm. (8)

These constraints are derived from prior knowledge, indicating that the patch dimensions within the unit cells are varied within the range of 0.05 mm to 7 mm. Similarly, for the unit cell with a through hole, the radius is varied within the range of 0.05 mm to 0.35 mm. It’s important to note that these parameter limits have been set with careful consideration of the maximum side length of the square unit cell, which is 7.5 mm, to prevent any overlap issues and ensure that the optimized metasurface design remains physically feasible and practical for fabricating a realistic and achievable design.”

Comment # 3:

In accordance with data presented in Fig.12 the goal of optimization has not been achieved due to a sidelobe at -20 degrees. The authors believe that this is the best possible result for the selected elements' geometry? Or was there no further optimization carried out due to time constraints?

Authors’ Response:

To ensure that the optimizer reaches the desired goal, we deliberately increased the requirement to -20 dB. However, the structure performs well even at -15dB. If we wish to achieve performance below -20dB, we can adjust the fitness function to push the sidelobe level (SLL) even lower, perhaps below -25dB. This can only be achieved to a certain extent without fundamentally challenging the underlying physics. The inherent periodicity in the system leads to the presence of grating lobes, which is a fundamental characteristic. Through this optimization approach, we can minimize grating lobes and sidelobes to a certain extent, but we cannot completely circumvent the physical limitations.

This approach can be likened to a black-box optimization, where we focus solely on the numerical optimization process without directly considering the underlying physics.

We conduct a guided search within the parameter space to find best possible outcome for a given requirement. The achieved results represent solutions that are near-optimal, without compromising the structural integrity or disrupting the underlying physics. Furthermore, this approach recognizes the trade-off between the achieved results and the time expended in the optimization process. Therefore, the stopping criterion was not fixed at FF = 0, where the fitness value reaches its absolute minimum. The definition of the stopping criterion also plays a role in determining the algorithm's convergence. In this instance, the optimizer halts when the diversity among the sampled candidates falls below a predefined threshold. At this point, the best parameters found are considered as the global optimum, which may be slightly different from the true global best but still effectively addresses the design problem under consideration.

Following text has been added to the revised manuscript on page 16, to address this point:

“The results obtained through this approach represent solutions that are in close proximity to the global optimum, while simultaneously preserving the structural integrity and not disrupting the underlying physics. Moreover, this method acknowledges the delicate balance between achieving desired results and the time invested in the optimization process. Consequently, the stopping criterion was not rigidly set to FF = 0, where the fitness value reaches its absolute minimum. The definition of the stopping criterion is a critical factor influencing the algorithm's convergence. In this particular case, the optimizer concludes its operation when the diversity among the sampled candidates in the Queue drops below a predefined threshold. At this juncture, the best parameters discovered are regarded as the global optimum. It's worth noting that this global optimum may differ slightly from the true global best, but it remains effective in addressing the specific design problem under consideration.”

Comment # 4:

The method proposed in this article (with the same simulation results) has already been presented by the authors earlier in DOI: 10.23919/URSI-EMTS.2019.8931531

Authors’ Response:

We apologize for any confusion, and we acknowledge that we should have provided clarification on this point earlier in the article. The optimization results presented in this study are very different from those in prior publication (DOI: 10.23919/URSI-EMTS.2019.8931531). In contrast to the periodic feed distribution briefly discussed previously in [10], the proposed approach assumes radial symmetry, which aligns more closely with the ideal amplitude scenario for feed tapering, such as Taylor taper or Chebyshev taper distribution.

A detailed explanation has been provided for this particular concern in response to Reviewer 1 (Comment 1) and appropriate changes have been made in the revised manuscript. Additionally, changes have been made in Fig. to represent the optimization approach correctly.

Reviewer 3 Report

1. what is the target of the figure .1 (a little bit more explaine relation between figure.1 and Algorithm 1)

2.  give more explanation for the drilled holes. ( reason, its effect of the impedance, S- PARAMETERS )

3. where are the properties of a unit cell? and compare it by super unit cell.

4. it is not clear for the reader the target of your structure in the text! is it based on gradient? is it reflectarray antenna? transmitarray?

5. why you consider multilayer structure? mentioen in the text.

5. 

Author Response

Reviewer: 3

Comment # 1:

What is the target of the figure .1 (a little bit more explained relation between figure.1 and Algorithm1)

Authors’ Response:

Figure 1 serves as a foundational representation of a model-based search algorithm. Given that the Cross-Entropy (CE) method relies on a pre-defined probability density function (model), the overall process can be succinctly characterized as an iterative sequence involving the definition of a model, sampling from that model, learning from the sampled data, and subsequently updating the initial model. For added clarity, we have included a reference that provides a comprehensive explanation of the model-based search method.

The following text is added on page 3 of the revised manuscript to enhance readability and ensure clarity for a broader audience.

“The CE optimization method follows a model-based search framework. In this approach, feasible solutions are derived from a parameterized probability distribution function (PDF), which is continually updated based on elite candidates identified in the previous iteration. Probability distributions are commonly referred to as "models" in the literature [25]. An elementary schematic diagram illustrating the CE search framework is presented in Fig.1. In essence, the Cross-Entropy (CE) method relies on a predefined PDF (model). The overall process can be concisely described as an iterative sequence encompassing the definition of a model, the sampling of data from that model, learning from the sampled data, and subsequently updating the initial model.”

Comment # 2:

Give more explanation for the drilled holes. (reason, its effect of the impedance, S- PARAMETERS).

Authors’ Response:

Thank you for highlighting this aspect. A detailed explanation of design methodology using holes in the PGM supercell is provided in a study previously conducted by the authors (DOI: 10.1109/ACCESS.2021.3100144). The effect on S-parameters has also been studied in details. The interested readers are referred to this article and following text has been added to reflect the same in the revised manuscript on page 7.

“Utilizing a unit cell with through holes extends the attainable phase range from a particular unit cell. For a more in-depth explanation, interested readers are encouraged to refer to [2], where a detailed elucidation on the design methodology for such PGMs is provided, offering enhanced clarity on the subject matter.”

Comment # 3:

where are the properties of a unit cell? and compare it by super unit cell.

Authors’ Response:

The properties of the unit cell are briefly discussed in the article on page 7 and the interested readers are referred to (DOI: 10.1109/ACCESS.2021.3100144) for further detailed on the entire design methodology of the PGM covering the unit cell, super cell and the full metasurface design. The manuscript has been revised to reflect these details and the exact texts are highlighted in red color in the revised manuscript on page 7.

“For a more in-depth explanation, interested readers are encouraged to refer to [2], where a detailed elucidation on the design methodology for such PGMs is provided, offering enhanced clarity on the subject matter.”

Comment # 4:

it is not clear for the reader the target of your structure in the text! is it based on gradient? is it reflectarray antenna? transmitarray?

Authors’ Response:

We are confident that by addressing the concerns raised by the reviewers, we have not only provided clarity regarding the design details but also clearly emphasized in the revised manuscript that the proposed structure is indeed a phase gradient metasurface.

Comment # 5:

why you consider multilayer structure? mention in the text.

Authors’ Response:

In response this comment, following text has been added to the manuscript on page 7. Other details are also added in response to the previous comments from the reviewer.

“The need for a multi-layer unit element 250 is driven by the essential requirement of achieving a 360â—¦ phase range, a crucial aspect 251 in the design of a Phase Gradient Metasurface (PGM), as elucidated in [ 1,2].”

Reviewer 4 Report

The article well written and organized.

However, there are several minor flaws listed below.

1) Line 5& Add explanation of PGM.

2) Lines 194-204: Check the first appearance and explanation of PTC and PGM, they have duplicates.

3) Text between lines 204 and 205, has no line numbers, check document layout.

4) There is repetition after line 231.

5) After expression (9) add explanation of T.

6) Text between lines 245 and 246, has no line numbers, check document layout.

7) Line 392 Add space after dot.

I refer to some questions for authors:

1. How can you estimate the effectiveness of the proposed method with typically used in CST (or other simulation tools)?

2. For the PGM dimension optimization, you used smoothing coefficient equals 0.2, which provides you finding optimal result at cost of ~448 hours computation time. Can you provide some estimation/suggestion about the influence of the smoothing coefficient on computation time and the criterion on the chose of the compromise value of smoothing coefficient?

3. Since the size of the unit cells was constant (half wavelength) with various sizes of metallic patches, I suppose that discretization of the wavefront is also not changing. Thus, it seems reasonable to consider cells or supercell as a Floquet cell and reformulate the optimization task in terms of magnitude and angle of S21-parameter. Seems like this allow to significantly reduce the whole computation time. Were authors consider that way? If so, please explain why it was rejected.

I suppose that to perform comparison and analysis may be considered another simplified and/or electromagnetically smaller model. If this work is already done, please provide the reference directly.

Author Response

Reviewer: 4

Comment # 1:

Lines 194-204: Check the first appearance and explanation of PTC and PGM, they have duplicates.

Authors’ Response:

Explanation regarding PGM is now added to the revised manuscript on page 7 while addressing the comments from Reviewer 3.

Comment # 2:

Lines 194-204: Check the first appearance and explanation of PTC and PGM, they have duplicates

Authors’ Response:

We are extremely thankful to the reviewer for directing our attention to the structuring error in the manuscript. We have now changed the arrangement and shifted the details of metasurface to the subsection “Periodic Metasurface Optimization Methodology”. Following text has been added in the revised manuscript on page 6:

“A periodic phase-gradient metasurface is shown in Fig. 2. These metasurfaces entail
an electrically expansive planar configuration consisting of repeating supercells arranged along both the x- and y-axes. This arrangement orchestrates a progressive phase modulation
in the electric field at the output, facilitating controlled beam tilting for the antenna’s operation. Because of the absence of a precise analytical model for metasurfaces, their effective optimization relies on employing a comprehensive electromagnetic simulation model.”

Also, the following repetitive text has been removed from the sub-subsection “Optimizing the Patch Dimension in PGM to Control SLLs” and added to subsection “Periodic Metasurface Optimization Methodology” for more clarity and better flow.

“We construct a 1-D metasurface, which corresponds to the highlighted strip in red within Fig. 3, in CST-MWS time-domain solver and defined appropriate boundary conditions to model the response of full metasurface ("E(t) = 0" along y-axis and the "Open Add Space" boundary condition along x-axis, as depicted in Fig. 2. By implementing these boundary conditions, the equivalent model offers a close approximation of the actual metasurface configuration while significantly reducing computational demands. With the supercell (comprising six individual cells) duplicated along the x-axis, the design parameters maintain the same periodic arrangement.”

The exact text added on page 7 to subsection “Periodic Metasurface Optimization Methodology” is as below:

“We construct an equivalent 1-D metasurface model, which corresponds to the highlighted section in red within Fig. 2. It has the same x-axis dimension (L = 345 mm) as the original metasurface aperture and y-axis dimension (W = 7.5 mm) as small as the dimension of the constituent unit cell. Appropriate boundary conditions are assigned to the 1-D metasurface model in CST-MWS time-domain solver to emulate the response of full metasurface ("E(t) = 0" along y-axis and the "Open Add Space" boundary condition along x-axis, as depicted in Fig. 3.”

Comment # 3:

Text between lines 204 and 205, has no line numbers, check document layout.

Authors’ Response:

This concern has been effectively addressed in response to the comment made by this reviewer. Additionally, it is plausible that the issue with line numbers was a technical matter on the MDPI system's end, and it is likely that they will resolve it if such issues persist

Comment # 4:

There is repetition after line 231.

Authors’ Response:

Yes, the line number have disappeared again after line 231. It is plausible that the issue with line numbers was a technical matter on the MDPI system's end, and it is likely that they will resolve it if such issues persist. In the revised manuscript the line numbers appear to be fine.

Comment # 5:

After expression (9) add explanation of T.

Authors’ Response:

We have added the following text to manuscript to address this issue:

“T stands for transpose”

Comment # 6:

Text between lines 245 and 246, has no line numbers, check document layout.

Authors’ Response:

Yes, the line number have disappeared again between line number 245 and 246. It is plausible that the issue with line numbers was a technical matter on the MDPI system's end, and it is likely that they will resolve it if such issues persist. In the revised manuscript the line numbers appear to be fine.

Comment # 7:

Line 392 Add space after dot.

Authors’ Response:

Thanks. We have made this correction in the revised manuscript.

Questions # 1:

How can you estimate the effectiveness of the proposed method with typically used in CST (or other simulation tools)?

Authors’ Response:

We do understand that it is extremely important to explain the effectiveness of the proposed method compared to the ones typically used in CST or other simulation tools. A detailed explanation has been provided as a response to a similar concern raised by Reviewer 1 in Comment 1 and corresponding revisions have been made to the revised manuscript which are highlighted in texts written in red color font.

Questions # 2:

For the PGM dimension optimization, you used smoothing coefficient equals 0.2, which provides you finding optimal result at cost of ~448 hours computation time. Can you provide some estimation/suggestion about the influence of the smoothing coefficient on computation time and the criterion on the close of the compromise value of smoothing coefficient?

Authors’ Response:

We express our gratitude to the reviewer for their comprehensive insights and thorough review, which have greatly assisted us in refining the article to make it more accessible and comprehensible to readers. The smoothing coefficient, denoted as alpha, plays a pivotal role in influencing the convergence of the algorithm. While a higher alpha value results in faster convergence, it does not guarantee outcomes that are necessarily near-optimal or optimal.

Before deciding on the final optimization parameters for the metasurfaces, we conducted a thorough examination of the CE algorithm's performance on a selection of benchmark problems. During this assessment, we experimented with various values of the smoothing parameter, namely 0.2, 0.5, and 0.7, using a 2D Ackley test function. Notably, for the 0.2 smoothing parameter, the number of iterations was higher, but it ultimately led to the discovery of the global best solution remarkably close to the peak of the search space.

Indeed, based on the information provided, it can be inferred that raising the value of the smoothing parameter will accelerate the convergence of the algorithm. However, this expedited convergence comes at the cost of the optimization outcomes potentially deviating from the global optimum. Therefore, there exists a tradeoff between the accuracy of attaining the global best solution and the computational time required for the algorithm to reach this global optimum. Balancing these factors is crucial when applying the algorithm to real-world problems, as the choice of the smoothing parameter can significantly impact the tradeoff between accuracy and computational efficiency.

To provide further insight, below is a snapshot illustrating the evolution of CE optimization process over the generations when applied to the 2D Ackley function when the smoothing parameter was set to 0.2. Conversely, for smoothing parameters of 0.5 and 0.7, convergence was achieved after four generations, but the global optimum diverged from the minimum point.

Questions # 3:

Since the size of the unit cells was constant (half wavelength) with various sizes of metallic patches, I suppose that discretization of the wavefront is also not changing. Thus, it seems reasonable to consider cells or supercell as a Floquet cell and reformulate the optimization task in terms of magnitude and angle of S21-parameter. Seems like this allow to significantly reduce the whole computation time. Were authors consider that way? If so, please explain why it was rejected.

Authors’ Response:

Thanks for highlighting this aspect. Indeed, smaller equivalent models, such as supercells with periodic boundary conditions, offer the advantage of reduced computation time while still providing fairly accurate predictions of metasurface performance. However, the effectiveness of this approach depends on the actual size of the metasurface and the number of times the supercell is repeated within the finite-sized metasurface. When the repetitions are limited to fewer than five, the results obtained using periodic boundary conditions may lack precision. The authors have previously explored these considerations in their published articles (reference). It should be noted that edge effects can introduce disruptions to phase and amplitude distributions when the metasurface contains fewer than eight supercell repetitions along the x-direction.

Hence, for electrically large apertures and practical metasurface designs, the proposed approach is most effective and reliably predicts the performance of the full metasurface. This crucial aspect has been underscored and emphasized in the revised manuscript:

“In contrast to the previously proposed PGM optimization strategies, as documented in Singh et al.'s works [1,2], which rely on a simplified supercell model and exclusively exploit surface periodicity, the method presented here employs a more precise equivalent model to anticipate the performance of finite-sized metasurfaces. The proposed optimization approach relies solely on the radiation pattern of the 1D metasurface, forecasted after conducting full-wave electromagnetic simulations, treating the problem as a black-box optimization. This distinguishes it from the alternative methods outlined in Singh et al.'s publications [1, 2], which delve into the intricacies of the simulation model's physics.

The outcomes projected by our proposed optimization approach closely align with real-world performance since the equivalent model applies the periodicity assumption exclusively along the y-direction while maintaining the x-direction boundary conditions identical to the original metasurface. In the case of significantly large PGM apertures, the Floquet analysis-based optimization approaches using supercells can significantly reduce computation time. However, they do not accurately predict the far-field pattern of finite-sized PGMs where supercell repetitions are less. For PGMs with lesser supercell repetitions, the optimization approach proposed in this work offers more reliable predictions and accurate optimization results.”

Questions # 4:

I suppose that to perform comparison and analysis may be considered another simplified and/or electromagnetically smaller model. If this work is already done, please provide the reference directly.

Such approach has been presented and has been found to be effective only under certain circumstances when the number if supercell repeating is enough to represent periodicity along x and y direction.

Authors’ Response:

We would like to draw the reviewer's attention to our published manuscripts [1, 2] which elaborate on our optimization approach utilizing a supercell framework with Floquet port excitation and periodic boundary conditions. In this methodology, we specifically target propagating reflection and transmission modes. We assign a higher weight to the desired mode while assigning weights to the undesired modes based on their magnitudes (with greater undesired mode magnitudes receiving higher weights). The fitness function is consequently formulated as a combination of these objectives, ensuring that the desired mode is amplified to exceed -0.1 dB, while the undesired modes are attenuated to fall below -25 dB. This approach effectively suppresses undesired grating lobes and sidelobes. However, it is important to note that this optimization strategy excels in enhancing the performance of electrically large aperture metasurfaces but faces challenges when applied to finite-sized metasurfaces with fewer supercell repetitions. In such cases, the optimized design variables may not exhibit behaviour consistent with predictions derived from supercell simulations and array calculator implementations.

The text added in the revised manuscript in response to Question 3 from the reviewer answers this as well.

Round 2

Reviewer 2 Report

Accept in present form.

Author Response

The reviewer has agreed to accept the manuscript in its current form. No other revision is required.

We thank the reviewer for constructive feedback that helped to improve the manuscript.

Reviewer 3 Report

I accept this manuscript to be published as a paper.

Author Response

No revisions have been asked.

Thanks for accepting the paper after detailed revisions.

Reviewer 4 Report

Thanks for the detailed answers.

Author Response

The reviewer has not raised any other concerns and is satisfied with the current state of the manuscript. Thanks for reviewing my work and helping me improve it with your feedback.